# Self-management of chronic, non-communicable diseases in South Asian settings: A systematic mixed-studies review

**Faraz Siddiqui** [1] *, **Catherine Hewitt**[1,2], **Hannah Jennings**[1,3], **Karen Coales**[1], **Laraib Mazhar**[4], **Melanie Boeckmann** [5], **Najma Siddiqi** [1,3,6]

**1** Department of Health Sciences, Mental Health and Addictions Research Group, University of York, York, United Kingdom, **2** Department of Health Sciences, York Trials Unit, University of York, York, United Kingdom, **3** Hull York Medical School, York, United Kingdom, **4** Penn State University College of Medicine, Hershey, Pennsylvania, United States of America, **5** Department of Global Health, Institute of Public Health and Nursing Research, University of Bremen, Bremen, Germany, **6** Bradford District Care NHS foundation trust, Bradford, United Kingdom

* faraz.siddiqui@york.ac.uk

**Data Availability Statement:** All the data underlying our findings are included within the

## Abstract

Self-management is crucial in mitigating the impacts of a growing non-communicable disease (NCD) burden, particularly in Low and Middle-Income countries. What influences self-management in these settings, however, is poorly understood. We aimed to identify the determinants of self-management in the high NCD region of South Asia and explore how they influence self-management. A systematic mixed-studies review was conducted. Key electronic databases [MEDLINE (1946+), Embase (1974+), PsycInfo (1967+) and CINAHL (EBSCOhost)] in March 2022 (and updated in April 2023) were searched for studies on the self-management of four high-burden NCD groups: cardiovascular diseases, type 2 diabetes, chronic respiratory diseases and depression. Study characteristics and quantitative data were extracted using a structured template, and qualitative information was extracted using NVivo. Quality appraisal was done using the Mixed Methods Assessment Tool (MMAT). Quantitative findings were organised using the Commission on Social Determinants of Health (CSDH) framework and synthesised narratively, supported by effect direction plots. Qualitative findings were thematically synthesised. Both were integrated in a mixed synthesis. Forty-four studies (26 quantitative, 16 qualitative and 2 mixed-methods studies) were included, the majority of which were conducted in urban settings and among individuals with diabetes and cardiovascular diseases. Higher age, education, and income (structural determinants), health-related knowledge, social support and self-efficacy (psychosocial determinants), longer illness duration and physical comorbidity (biologic determinants), and the affordability of medicine (health-system determinants) were key determinants of self-management. Qualitative themes highlighted the role of financial adversity and the social and physical environment in shaping self-management. A complex interplay of structural and intermediary social determinants shapes self-management in South Asian settings. Multi-component, whole-systems approaches could boost self-management in these settings. Key areas include empowerment and education of patients and wider

manuscript and supplementary information provided.

**Funding:** The authors received no specific funding for this work.

**Competing interests:** I have read the journal's policy and the authors of this manuscript have the following competing interests: Najma Siddiqi is the Principal Investigator of the DiaDeM Global Health Research Program [Grant reference: Research & Innovation for Global Health Transformation (RIGHT) NIHR200806]. Catherine Hewitt and Hannah Jennings are co-investigators and leads for the DiaDeM Behavioural Activation trial and process evaluation work packages, respectively. The authors do not have any other competing interests to declare.

community, design and delivery of bespoke behavioural interventions and a stronger emphasis on supporting self-management in healthcare settings.

## Introduction

Non-communicable diseases (NCDs) encompass a range of chronic physical and mental health conditions that have witnessed an exponential rise globally in the past three decades. Attributed to a pool of shared risk factors arising from global development [1], NCDs disproportionately affect populations in Low and Middle-Income Countries (LMIC), both in terms of disease risk and sequelae. Populations living in LMIC regions account for 85% of global premature mortality (death between 30–69 years) and suffer the greatest economic impacts from NCDs [2,3]. South Asia is of particular importance in this regard—it is a densely populated LMIC region with a high prevalence of NCDs, particularly cardiometabolic diseases, respiratory diseases and depression [4,5]. Underlying populations in this region are similar demographically and share common biological, social and environmental influences that shape illness experiences [6].

Tackling NCDs requires a multi-pronged approach, a key component of which is self-management through activities such as exercise, dietary regulation, medication adherence, risk reduction and the use of appropriate coping strategies that may be undertaken with healthcare provider support [7,8]. In high-income countries, considerable effort has been invested in understanding how individuals self-manage chronic NCDs and in supporting self-management to improve illness control and overall health outcomes [9]. The focus of these efforts is towards empowering individuals with NCDs, promoting a collaborative model of care in which the individuals are fully engaged in managing their illness. Such a paradigm shift is a key to overcoming challenges faced by ill-equipped and overburdened health systems, which are characteristic traits of South Asian and wider LMIC regions [10].

Improving self-management in a group of individuals requires context-specific knowledge. This is particularly relevant for South Asian settings and populations, which are characterised by inefficient and poorly resourced health systems [11], deep-rooted socio-cultural practices and traditions [12], as well culturally ingrained beliefs [13] and practices around health [14]. From existing evidence, we know that self-management of NCDs in South Asian and wider LMIC populations is sub-optimal [15,16]. With evidence largely coming from studies conducted in non-South Asian populations [17,18], there is much less known about the influences that individual and wider socio-cultural factors highlighted above have on self-management in these settings. To inform the development, planning and implementation of contextually appropriate interventions, an in-depth exploration of these factors with a focus on the South Asian perspective is needed. The current study addresses this knowledge gap through a comprehensive synthesis of current evidence. It aims to identify the determinants that influence self-management and their mechanisms across a set of commonly occurring NCDs in South Asian settings.

## Methods

We conducted a systematic mixed-studies review, using a convergent segregated approach to undertake independent quantitative and qualitative syntheses, which were then integrated in in the discussion [19]. The review was registered prospectively with PROSPERO

(CRD42021240899, amended to focus on South Asia rather than on LMICs) and is reported in line with PRISMA [20] guidelines (S1 Table).

## Eligibility criteria

We included primary quantitative, qualitative and mixed-methods studies on the self-management of four high-burden diseases with common underlying risk factors: cardiovascular diseases (angina, heart failure, hypertension, stroke), chronic respiratory diseases (asthma and COPD), type 2 diabetes mellitus and depression [6,21]. Studies conducted among adults (≥18 years) in a single or multiple South Asian countries were eligible. Quantitative studies that reported associations of identified determinants with overall self-management, or with either one or more individually observed self-management behaviours were included. We used the American Academy of Diabetes Educators' (AADE) list of self-management behaviours for diabetes, which by extension is also relevant to the other NCDs included in this review. These behaviours include: healthy coping, healthy eating, being active, taking medication, monitoring, reducing risk and problem solving [7]. Qualitative studies that explored self-management of the selected NCDs, or aimed to identify the facilitators and barriers to carrying out self-management activities were included. Mixed-methods studies that fulfilled either criterion for quantitative or qualitative studies were included.

## Search strategy

An initial scoping search of studies conducted in LMICs was carried out to identify studies conducted in South Asia. The scoping search, pilot searches and feedback from an information specialist at the University of York informed the full search strategy (S1 Text)- it was based on three key concepts (self-management, NCDs, and LMICs) and used a combination of free text and subject headings, as well as appropriate search functions and the Cochrane Effective Practice and Care (EPOC) search filter for LMICs. The original search was carried out in March 2022 on four electronic databases and updated in April 2023. The following databases were searched: MEDLINE (Ovid, Epub Ahead of Print, In-Process, In-Data-Review & Other Non-Indexed Citations and Daily, 1946+), Embase (Ovid, 1974+), APA PsycInfo (Ovid, 1967+) and CINAHL (EBSCOhost). We additionally searched the grey literature (Opengrey and Proquest) and references of relevant articles to identify potentially eligible reports that were missed in the electronic search.

## Study selection and data extraction

Title and abstract (first stage) screening were carried out using Rayyan software to identify potentially relevant articles. We used the World Bank country classification for South Asia (a list of eight countries which comprises Afghanistan, Bangladesh, Bhutan, India, Maldives, Nepal, Pakistan and Sri Lanka) to identify South Asian studies which were then screened for eligibility [22]. Included references were transferred to Endnote which was used for the retrieval and full-text (second stage) screening. Both sets of screening were independently undertaken by two reviewers, and any disagreement was resolved through discussion.

A structured, pre-tested electronic data extraction form was used to collect information on study characteristics, participants, methods and contextual information, key findings and strengths and weaknesses for each study. Information relevant to quantitative and qualitative synthesis was extracted separately [23]. These included observed measures of association (ORs and 95% CIs, or equivalent) which were extracted using an extension of the same data extraction form. The extraction and synthesis of qualitative findings were carried out using NVivo,

version 12 [24]. Key findings relating to self-management and participant quotes supporting these were identified from full texts. They were then coded and organised for synthesis.

### Quality assessment

Two independent researchers assessed the quality of each study using checklists specific to quantitative, qualitative or mixed methods studies in the Mixed Methods Assessment Tool (MMAT) [25]. Responses to each assessment item were recorded as "yes", "no" or "can't tell", with one point assigned for each item scoring as "yes". Overall study scores were used to identify studies as low (0–1 points), moderate (2–3 points) or high quality ($\geq$4 points).

### Evidence synthesis

The associations between identified determinants and self-management were synthesised narratively in view of the methodological and clinical heterogeneity across studies, and then reported in line with the synthesis without meta-analysis (SWIM) guidelines [26]. We used the World Health Organisation's Commission on Social Determinants of Health (CSDH) framework to identify determinants that act at the structural level (socio-economic and political context, indicators of socioeconomic position) and the intermediary level (material circumstances, behavioural and biological factors, psychosocial factors, health-system related factors) [27]. Descriptive summaries were produced initially for each determinant in a preliminary synthesis describing the observed direction of associations, patterns across studies, and the quality of the overall evidence. These were supported by the use of effect direction plots [28], which provided a visual representation of reported associations and key study characteristics, i.e. study quality, study size, type of self-management behaviour and NCDs. The findings were presented based on the strength of the evidence, using study quality as a guide.

Qualitative findings were synthesised inductively using a thematic approach [29]. The synthesis process was undertaken in three steps. In the first step, Illustrative quotes (first-order constructs) and authors' interpretations (second-order constructs) relevant to the process of self-management, or its barriers and facilitators were coded. The list of codes generated was then reviewed, refined and organised to form descriptive themes. In the last step, descriptive themes were synthesised to develop overarching analytic themes (third-order constructs).

The quantitative and qualitative syntheses were integrated through cross-comparison and linking of findings reported in each synthesis, as recommended by published guidance [19]. This was undertaken by exploring the extent to which individual syntheses were complementary or contradictory, observing what qualitative insights were added for the observed quantitative associations and identifying shortcomings in each individual synthesis to reflect critically on how these were addressed through integration. The explanations and contextual detail supporting the explored ideas were then summarised.

## Results

A total of 22,112 studies were identified through electronic database searches (n = 22,089) and searches of reference and grey literature (n = 23). 1,656 titles and abstracts were screened after deduplication and removal of non-South Asian studies. Of these, 64 proceeded to full-text screening and 44 were ultimately included in the review (Fig 1). Studies were predominantly conducted in India (n = 19) and in urban (n = 30) settings. The included studies were mostly conducted in individuals with type 2 diabetes (n = 21), followed by cardiovascular (n = 18) and chronic respiratory diseases (n = 5). Co-morbid status was recorded in several studies, but one study (Yadav et. al, 2020a) exclusively sampled patients with multimorbidity (Table 1).

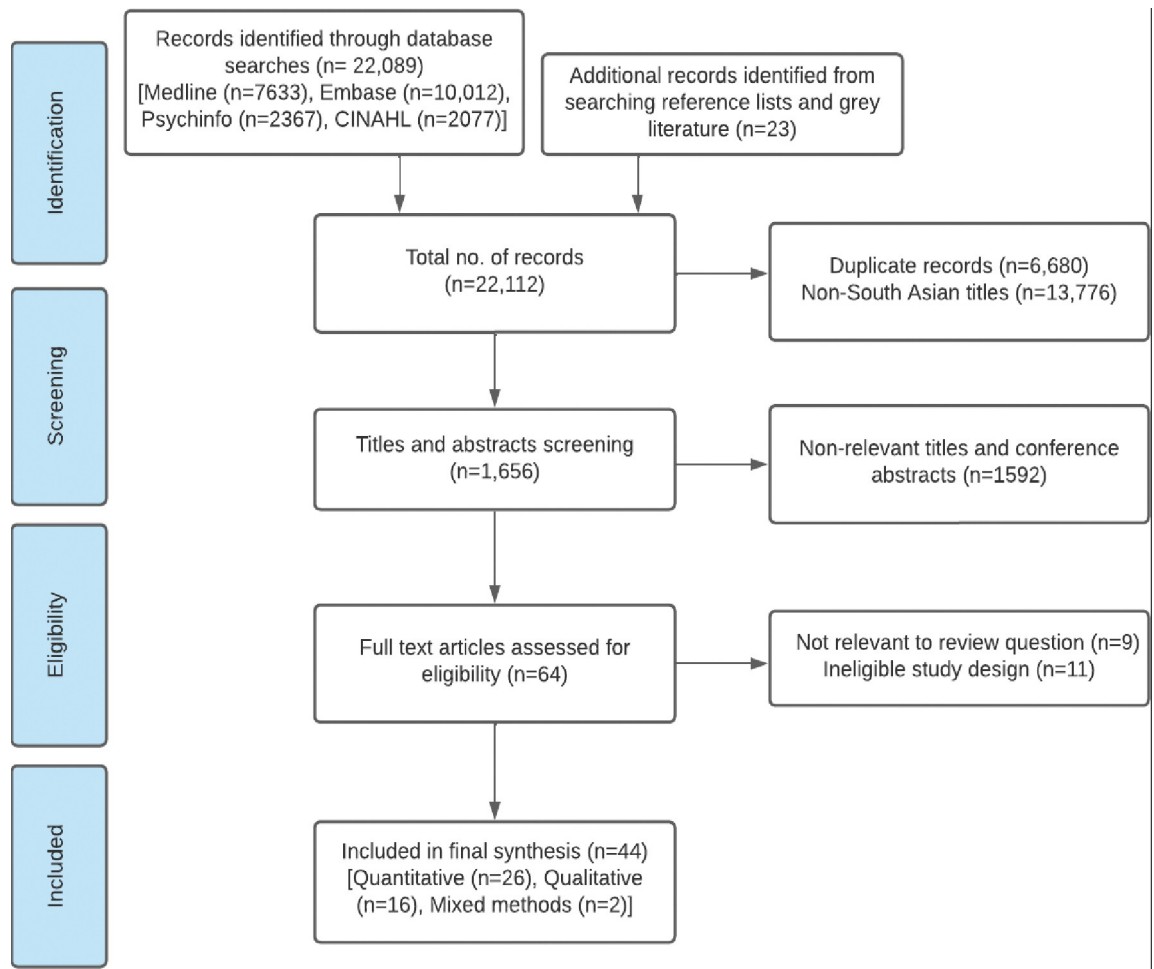

**Fig 1. Overview of the study selection process–the PRISMA flow diagram.**

Quantitative studies (n = 26) were all cross-sectional, qualitative studies (n = 16) were predominantly descriptive (n = 13); the two mixed methods studies were of concurrent and sequential design. All included studies were published in the last ten years. Around two-thirds of all included studies were of moderate (n = 25) or low (n = 9) quality. Higher quality scores were observed for qualitative and mixed-methods studies. (S2 Table).

## Quantitative synthesis

Across the included studies, commonly observed indicators of **socioeconomic position** included: age (n = 13), education (n = 15), gender (n = 16), individua/household income (n = 11) and occupation (n = 5). Comorbidity (n = 11) and illness-related factors (i.e., illness duration, control over illness, and treatment complexity) (n = 14) were commonly observed **behavioural and biological** determinants. Health-related knowledge (n = 5), social support (n = 10) and attitudes and beliefs (n = 4) were the main **psychosocial determinants** identified. Access to affordable treatment (n = 4) and factors related to the healthcare provider (type of provider, regularity of follow-up and treatment satisfaction) (n = 6) represented **health-system-related determinants** (Table 2).

**Table 1. Characteristics of primary quantitative, qualitative and mixed-methods studies included in the review (n = 44).**

| Author, year | Study design | Study objectives | Study settings and location(s) | Study population | Participant characteristics* | Data collection and analysis methods |
|---|---|---|---|---|---|---|
| Aggarwal, 2017 [30] | Quantitative, cross-sectional | To assess the determinants of adherence to inhaled corticosteroids among asthmatic patients | Single-center, tertiary care setting in Chandigarh, India (Urban) | Individuals with stable asthma, on treatment for >6 months | 103 participants; 41.9±16 years; 43 (41.7%) males | Self-reported medication adherence; individuals taking ≥80% of prescribed dosage identified as adherent. Logistic regression used to identify predictors of adherence |
| Ajani, 2021 [31] | Quantitative, cross-sectional | To report the level of self-care and its predictors among hypertensive patients | Multi-center, tertiary care settings in Karachi, Pakistan (Urban) | Individuals (≥18 years) with hypertension and on anti-hypertensive treatment | 402 participants; median: 55 years (range: 48–65); 136 (33.8%) males | Multiple self-management behaviours (medication adherence, diet, physical activity, weight control, smoking cessation) assessed using H-scale. Logistic regression was used to identify predictors of adherence after dichotomising scores (cut-off not provided). |
| Arulmozhi, 2014 [32] | Quantitative, cross-sectional | To assess medication adherence and adherence to self-care among T2DM patients | Single-center, tertiary care setting in Puducherry, India (Rural) | Individuals (In-patients) with T2DM >1 month irrespective of diagnosis at admission. | 150 participants; 54±12 years; 75 (50%) males | Medication adherence was measured using the MMAS and categorised as low (<6), medium (6–7) or high (≥8). Logistic regression used to identify predictors of high medication adherence |
| Basu, 2018 [33] | Quantitative, cross-sectional | To assess adherence to self-care practices including medication intake and influencing factors among T2DM patients | Single-center, tertiary care setting in Delhi, India (Urban) | Individuals (18–65 years) with T2DM on treatment for ≥1 year | 375 participants; 49.7±10 years; 201 (53.6%) males | Medication adherence measured using the SDSCA scale. Adherence was defined as missed medications on ≤1 day in the past week). Logistic regression was used to identify predictors of medication non-adherence. |
| Bhandari B, 2015 [34] | Quantitative, cross-sectional | To explore adherence to antihypertensive treatment and identify factors associated with treatment non-adherence | Single-center, community-based setting in Dharan, Nepal (Urban) | Individuals (>35 years) with hypertension diagnosed in the past year. | 154 participants; 77 (50.0%) <55 years; 71 (46.1%) males | Medication adherence using the 4-item MMAS. Adherence was defined as an MMAS score <3. Logistic regression used to identify predictors of medication non-adherence |
| Bhandari S, 2015 [35] | Quantitative, cross-sectional | To identify the prevalence and predictors of antihypertensive medication adherence | Single-center, community-based setting in Kolkata, India (Urban) | Individuals (≥25 years) with hypertension, currently on antihypertensive treatment and able to speak Bengali | 348 participants; 55.1±12.3 years; 111 (32.0%) males | Self-reported medication adherence—those taking ≥80% of prescribed dosage identified as adherent. Logistic regression used to identify predictors of adherence |
| Chandrika, 2020 [36] | Quantitative, cross-sectional | To assess self-care behaviours and its associated factors among people with T2DM | Single-center, community-based setting in Hyderabad, India (Urban) | Individuals (>18 years) with T2DM willing to give written consent | 208 participants; 51.3 ±9.4 years; 95 (45%) males | Multiple self-management behaviours (diet, exercise, self-monitoring, drug adherence), using the modified SDSCA scale Logistic regression used to identify predictors of good dietary, exercise, self-monitoring behaviours and drug adherence |

*(Continued)*

**Table 1.** (Continued)

| Author, year | Study design | Study objectives | Study settings and location(s) | Study population | Participant characteristics* | Data collection and analysis methods |
|---|---|---|---|---|---|---|
| Ghimire, 2017 [37] | Quantitative, cross-sectional | To identify the psychosocial barriers encountered by T2DM patients in achieving recommended diet and exercise | Single-center, tertiary care setting in Kupondole, Nepal (Urban) | individuals with T2DM (≥18 years) who received consultant advice on diet and exercise | 197 participants; 54.7±11.3 years; 111 (56.3%) males | Multiple self-management behaviours (diet, physical activity). Compliance was defined as physician recommended diet for ≥6 days/week (diet) and ≥5 sessions of moderate or vigorous activity per week (physical activity) Logistic regression was used to identify predictors of non-compliance to diet and physical activity |
| Ghimire, 2018 [38] | Quantitative, cross-sectional | To identify the barriers to dietary salt reduction among hypertensive patients | Single-center, tertiary care setting in Kathmandu, Nepal (Urban) | Individuals with hypertension who received consultant advice on dietary salt reduction ≥ 12 weeks ago. | 180 participants; 52.4±13 years (compliant), 54±12.1 years (non-compliant); 90 (50%) males | Dietary behaviour (salt reduction) measured using patients' self-report of compliance to physician's recommendations and reduced dietary salt intake. Logistic regression was used to identify predictors of non-compliance. |
| Gopichandran, 2012 [39] | Quantitative, cross-sectional | To estimate self-care behaviours and factors influencing these behaviours in T2DM | Single-center, community-based setting in Vellore, South India (Urban) | Individuals (>18 years) with T2DM, independently carrying out activities of daily living. | 200 participants; 16 (8%) <40 years, 58 (29%) 41–50 years, 66 (33%) 51–60 years, 41 (20.5%) 61–70 years: 19 (9.5%) 71–80 years; 82 (41%) males | Multiple self-management behaviours (diet, exercise, drug adherence) measured using the SDSCA scale. Logistic regression was used to identify the predictors of good dietary and exercise behaviours and drug adherence. |
| Gowani, 2017 (a) [40] | Quantitative, cross-sectional | To explore self-care behaviour and its association with socio-demographic and clinical factors in patients living with HF | Multi-center, tertiary care settings in Karachi, Pakistan (Urban) | Individuals (in-patients) with a diagnosis of HF (LVEF<45%) and residing in Karachi | 230 participants; 58.4±12.1 years; 152 (66.1%) males | Self-care behaviour assessed using 9 questions of the EHFSCB tool, with lower scores indicating better self-care. Linear regression was used to identify predictors of self-care behaviour. |
| Kumar Gupta, 2022 [41] | Quantitative, cross-sectional | to ascertain foot self-care behaviour and its associated factors among people with T2DM | Single-center, community-based setting in Punjab, India (rural) | Individuals (≥30 years) with T2DM taking antidiabetic drugs for >6 months | 700 participants; 56 (50–62) years [median (IQR)]; 233 (33%) males | Foot self-care assessed using the NAFF questionnaire scores and categorised as good (>70%), satisfactory (50–70%) or poor (<50%) Logistic regression analysis was used to identify predictors of foot self-care practices |
| Kandel, 2022 [42] | Quantitative, cross-sectional | to assess the association between behaviours of family members and self-care activities of patients with T2DM | Single-center, tertiary care setting in Kathmandu, Nepal (urban) | Individuals (≥18 years) with T2DM for ≥3 months and able to communicate in Nepali | 411 participants; 47 (11.4%) 21–35 years, 119 (28.9%) 36–50 years, 177 (43.1%) 51–65 years, 68 (16.6%) >65 years; 177 (43.0%) males | Overall self-management based on the five domains (medication adherence, physical activity, self-monitoring, dietary behaviour, and foot care) using the SDSCA. Logistic regression analysis with Bonferroni correction used to identify the association between family behaviours and T2DM self-management |

*(Continued)*

**Table 1.** (Continued)

| Author, year | Study design | Study objectives | Study settings and location(s) | Study population | Participant characteristics* | Data collection and analysis methods |
|---|---|---|---|---|---|---|
| Khanam, 2014 [43] | Quantitative, cross-sectional | To describe hypertension and the determinants of non-adherence to treatment among adult hypertensive persons | Multi-center, community-based settings in Matlab, Abhobyagar and Mirsarai—Bangladesh (Rural) | Individuals (≥25 years) with hypertension and residing in the three catchment sites | 4097 participants; 44.6±15.8 years; 47.4% males | Medication adherence, based on self-reported use of medication at the time of interview. Logistic regression used to identify predictors of medication non-adherence |
| Koirala, 2020 [44] | Quantitative, cross-sectional | To determine the context of self-care and examine sociodemographic and clinical factors affecting self-care | Multi-center, tertiary care settings in Kathmandu, Nepal (Urban) | Individuals (Inpatients) with a diagnosis of HF ≥1 month; able to understand Nepali and provide consent | 221 participants; 57.5±15.7 years; 136 (61.8%) males | Self-care maintenance scores measured using the SCHFI subscales. Linear regression used to identify predictors of self-care maintenance |
| Mahmood, 2020 [45] | Quantitative, cross-sectional | To evaluate and compare the degree of medication adherence in hypertensive patients in primary, secondary and tertiary care settings | Multi-center, primary, secondary and tertiary care settings in Islamabad, Pakistan (Rural) | Individuals (≥18 years) with a diagnosis of hypertension able to converse in Urdu | 741 participants; 53.6±12.6 years, 389 (52.5%) males | Medication adherence measured using the 8-item MMAS and categorised as adherent (8), moderately adherent (6–7) or non-adherent (<6). Logistic regression used to identify predictors of medication adherence |
| Mannan, 2021 [46] | Quantitative, cross-sectional | To determine the levels of medication adherence in patients with T2DM and analyze the factors associated with poor adherence | Multi-center, tertiary care settings in Chittagong, Bangladesh (Urban) | Individuals with T2DM on oral medications, residing in Chittagong | 2061 participants; 50.6±12.1 years, 1233 (59.8%) males | Medication adherence measured using the 8-item MMAS, and categorised as adherent (8), moderately adherent (6–7) or non-adherent (<6). Logistic regression used to identify predictors of low medication adherence (vs medium and high) |
| Saqlain, 2019 [47] | Quantitative, cross-sectional | To assess medication adherence and its association with socio-demographic, health-related, patient-related and disease-related characteristics among hypertensive individuals | Single-center, tertiary care setting in Islamabad, Pakistan (Urban) | individuals (>65 years) with a diagnosis of hypertension, taking ≥1 medication for the past month | 262 participants; 22 (84.7%) 65–75 years, 29 (11.1%) 76–85 years, 11 (4.2%) >85 years; 93 (35.4%) males | Medication adherence measured using the 4-item MAQ, adherence defined as a score of 4. Logistic regression analysis used to identify predictors of medication adherence |
| Sirari, 2019 [48] | Quantitative, cross-sectional | To measure the compliance level of self-care practices among T2DM patients | Single-center, tertiary care setting in India (Urban) | Individuals (>18 years) with T2DM registered at the healthcare facility | 60 participants; 50% males; 54.9±9.2 years | Multiple self-management behaviours (diet, physical activity, and foot care) assessed using the SDSCA scale. Logistic regression used to identify predictors of adherence to each of the above behaviours |
| Shani, 2021 [49] | Quantitative, cross-sectional | To identify the facilitators and barriers to medication adherence among survivors of the first episode of stroke | Single-center, tertiary care setting in India (Urban) | Individuals (≥18 years) who had a primary stroke within a period of three months to one year | 240 participants; 31 (12.9%) ≤45 years, 209 (87.1%) >45 years; 179 (74.6%) males | Self-reported medication adherence—those taking ≥80% of prescribed dosage identified as adherent. Logistic regression used to identify predictors of medication adherence |

(*Continued*)

**Table 1.** (Continued)

| Author, year | Study design | Study objectives | Study settings and location(s) | Study population | Participant characteristics* | Data collection and analysis methods |
|---|---|---|---|---|---|---|
| Shrestha, 2021 [50] | Quantitative, cross-sectional | To assess the association between subthreshold depression and self-care behaviours in adults with type 2 diabetes | Single-center, tertiary care setting in Kathmandu, Nepal (Urban) | Individuals with T2DM since ≥1 year, fluent in Nepalese | 354 participants; 51.6±12.5 years; 156 (44%) males | Overall and individual self-management behaviours (diet, physical activity, medication adherence, foot care) assessed using the SDSCA. Linear and logistic regression were used to identify the association of subthreshold depression with overall and individual self-management behaviours, respectively |
| Rao, 2014 [51] | Quantitative, cross-sectional | To determine the factors responsible for medication non-compliance in hypertensive and diabetic patients | Single-center, community-based setting in Karnataka, India (Rural) | Community-based individuals (≥30 years) previously identified as having hypertension or T2DM | 287 hypertensive, 139 diabetic, 80 hypertensive and diabetic participants (gender not provided) | Self-reported medication adherence—those taking ≥80% of prescribed dosage identified as adherent. Logistic regression analysis used to determine predictors of medication non-adherence |
| Rafi, 2022 [52] | Quantitative, cross-sectional | To assess adherence and factors associated with non-adherence to inhalers among asthma patients | Single-center, tertiary care setting in Rajshahi, Bangladesh (Urban) | Individuals (≥18 years) with asthma visiting the healthcare facility | 357 participants; 34.5 ±10.1 years; 232 (64.9%) males | Inhaler adherence measured using Test of Adherence scale; non-adherence defined as having a score ≤45. Logistic regression used to identify predictors of inhaler non-adherence |
| Ravi, 2018 [35] | Quantitative, cross-sectional | To assess the influence of family support on self-management of patients with diabetes | Single-center, tertiary care setting in Chennai, India (Urban) | Individuals (>30 years) with T2DM | 200 participants; 20 (10.1%) 31–40 years, 60 (30.2%) 41–50 years, 55 (27.6%) 51–60 years, 54 (27.1%) 61–70 years, 10 (5.0%) >70 years; 96 (48%) males | Self-management assessed using the SDSCA; Structural equation modelling used to identify the association between social support and self-management. |
| Roka, 2020 [53] | Quantitative, cross-sectional | To assess medication adherence and factors associated with low adherence among hypertensive patients | Single-center, tertiary care setting in Kathmandu, Nepal (Urban) | Individuals with a diagnosis of hypertension >6 months | 216 participants; 31 (14.4%) 31–50 years, 143 (80.6%) 51–70 years, 42 (19.4%) >70 years; 110 (50.9%) males | Medication adherence assessed using the MMAS (version not specified) Logistic regression used to identify predictors of low adherence |
| Yadav, 2020 (a) [54] | Quantitative, cross-sectional | To examine the level of self-management practice, and its relationship with socio-demographic factors, health literacy and patient activation | Multi-center, community-based settings in Sunsari district, Nepal (Rural) | Individuals (18–70 years) with COPD and ≥1 pre-specified co-morbidities: CVD, T2DM, asthma, arthritis, depression, Musculo-skeletal disorders or gastritis | 238 participants; 18±12.1 years; 107 (45%) males | Self-management practice scores assessed using the 18-item SMPQ. Lower scores indicated poor Self-management practice. Linear regression was used to identify the factors associated with self-management practice. |
| Adhikari, 2021 [55] | Qualitative, descriptive | To explore barriers to and facilitators of T2DM self-management practices from multiple stakeholder perspectives | Multi-center, primary and tertiary care settings in Rupandehi, Nepal (Rural) | Individuals (>20 years) with T2DM, caregivers, physicians, public health professionals, social workers | 26 T2DM patients; 5 caregivers, 7 physicians, 3 public health professionals and a social worker (gender not provided) | Four FGDs and 16 IDIs explored barriers and facilitators to Self-Management practices. Data were analysed thematically and organised using the socio-ecologic model. |

*(Continued)*

**Table 1.** (Continued)

| Author, year | Study design | Study objectives | Study settings and location(s) | Study population | Participant characteristics* | Data collection and analysis methods |
|---|---|---|---|---|---|---|
| Anitha Rani, 2019 [56] | Qualitative, descriptive | To determine perceptions of patients with T2DM about barriers to self-management | Single-center, secondary care setting in Chennai, India (Rural) | Individuals (>40 years) with T2DM, on treatment for ≥6 months | 50 participants; 58±10.5 years; (gender not provided) | Five FGDs explored aspects of self-care relating to diet, physical activity, foot care, medication adherence, complications, and target blood levels. Data were analysed thematically. |
| Ansari, 2021 [57] | Qualitative, descriptive | To explore HCPs' perspectives of T2DM patients' experiences of self-management | Multi-center, primary care settings in Abottabad, Pakistan (Rural) | HCPs (physicians and nurses) with ≥10 years in T2DM management | 20 HCPs- 10 physicians, 5 (50%) males, 10 female nurses | IDIs explored HCPs views on difficulties in T2DM self-management. Data were analysed thematically. |
| Ansari, 2019 [58] | Qualitative, descriptive | To explore the factors contributing to optimal self-management skills among middle-aged T2DM patients | Single-center, primary care settings in Abottabad, Pakistan (Rural) | Individuals with T2DM (40–60 years) and poorly controlled diabetes (HbA1C>7.0%) | 30 participants; 58 years; 15 (50%) males | IDIs collected data on knowledge and practices toward diabetes medicine, self-monitoring, healthy eating, physical activity, and continuity of care. Data were analysed thematically. |
| Chittem, 2021 [59] | Qualitative, descriptive | To explore the barriers to self-monitoring and medication management among T2DM patients, primary caregivers and HCPs | Multi-center, secondary care settings in India (Urban) | Individuals (≥18 years) with T2DM, their primary caregivers and physicians | 50 T2DM patients; 42.5 years; 36 (72%) males; 50 caregivers and 25 physicians | IDIs inquired into the main barriers the participant encountered in the practice of medication and self-monitoring. Thematic analysis was used to analyse data. |
| Basu, 2020 [60] | Qualitative, descriptive | To determine reasons for non-adherence to treatment and its socioeconomic determinants among low-income hypertensive patients. | Single-center, primary care settings in Delhi, India (Urban) | Individuals (35–65 years) with hypertension, on anti-hypertensive treatment for >2 years | 30 participants; 53.4±8.1 years; 16 (53.3%) males | IDIs collected patient perspectives on medical adherence to prescribed antihypertensive therapy and recommended lifestyle modifications. Data were analysed thematically. |
| Bukhsh, 2020 [61] | Qualitative, descriptive | To provide insights into the experiences, behaviours, and barriers to self-care practice among urban adults with T2DM. | Multi-center, secondary and tertiary care settings in Pakistan (Urban) | Individuals (>30 years) with T2DM, diagnosed >1 year ago | 37 participants; 54.8±9.6 years; 11 (34.5%) males | IDIs collected data on knowledge and practice towards diabetes medicine, self-monitoring, healthy eating, physical activity and continuity of care. Data were analysed thematically. |
| Gowani, 2017 (b) [62] | Qualitative, phenomenological | To explore the experience of patients living with Chronic HF. | Multi-center, tertiary care settings in Karachi, Pakistan (Urban) | Individuals (in-patients and outpatients) with HF identified via hospital records | 8 participants; range:38–72 years; 5 (62.5%) males | IDIs explored patients' experiences of living with HF, its impact on daily life and the experiences with the healthcare system for HF treatment. Data were analysed using steps identified by Morse and Niehaus (2007). |

(*Continued*)

**Table 1.** (Continued)

| Author, year | Study design | Study objectives | Study settings and location(s) | Study population | Participant characteristics* | Data collection and analysis methods |
|---|---|---|---|---|---|---|
| Gupta, 2019 [63] | Qualitative, descriptive | To evaluate the barriers and other determinants of hypertension awareness, treatment and control among hypertensive women | Single-center, tertiary care setting in Jaipur, India (Rural) | Females with hypertension, on anti-hypertensive treatment since ≥12 months | 30 participants; 56.5 + 8.9 years (all female) | Open-ended questions relating to medication adherence were administered using a semi-structured questionnaire followed by a free conversation. A descriptive analysis was undertaken. |
| Islam, 2017 [64] | Qualitative, descriptive | To understand patients' perspective on T2DM and factors underscoring their adherence to diabetes medications | Single-center, tertiary care setting in Dhaka, Bangladesh (Urban) | Individuals with T2DM since ≥1 year, taking oral hypoglycemic drugs | 12 participants; 52.2 years; 5 (41.6%) males | IDIs explored participants' knowledge and perception about diabetes and its long-term health impact; views on diabetes treatment and the importance of adherence to medications. Data were analysed using content analysis. |
| Jose, 2020 [65] | Qualitative, descriptive | To describe perceptions on the facilitators and barriers to HF care | Singe-center, tertiary care setting in Kerala, India (Urban) | Stakeholders of HF care, i.e., including patients, caregivers, and HCPs | FGDs: 17 participants—12 patients, 5 caregivers IDIs: 22 patients range:34–77 years; 19 (86.3%) males 13 HCPs, 9 caregivers (details not provided) | 14 IDIs with HCPs, patients, and caregivers, and 3 FGDs (n = 3) with patients and caregivers explored barriers to HF care. Data were analysed thematically. |
| Kamath, 2020 [66] | Qualitative, descriptive | To explore the perspective of self-care facilitators in HF. | Single-center, tertiary care setting in India (Urban) | Individuals (≥18 years) with HF (NYHA class II–IV, with ≥4 weeks before index hospitalization) and their caregivers | 15 participants- 8 patients; 7 caregivers; 60 ± 13.6 years; 4 (50%) males (patients only) | IDIs were conducted to explore aspects of monitoring, maintenance and management. Data were analysed using content analysis. |
| Kamath, 2021 [67] | Qualitative, grounded theory | To understand the factors affecting self-care among Chronic HF | Single-center, tertiary care setting in India (Urban) | Individuals (≥18 years) with HF (NYHA class II–IV, with ≥4 weeks before index hospitalization) and their caregivers. | 39 participants—22 patients; 17 caregivers; 61 + 13.4 years; 13 (59.1%) males | IDIs conducted with patients ascertained knowledge and self-management. Caregiver interviews focused on the role played by caregivers. Data were analysed using Chamaz's approach to grounded theory. |
| Matpady, 2020 [68] | Qualitative, descriptive | To explore knowledge, current dietary practices, and the barriers and enablers for dietary self-care management in T2DM | Multi-center settings Karnataka, India (Urban) | Individuals (30–65 years) with T2DM scoring lowest or highest on the Diabetes Self-management Questionnaire | 35 participants; range: 30–65 years; 20 (57%) males | IDIs focused on their experience in diabetes self-management practice. Data were analysed thematically. |
| Yadav, 2020 (b) [69] | Qualitative, descriptive | To understand the facilitators and barriers affecting COPD self-management | Multi-center, community-based settings in Sunsari district, Nepal (Rural) | Individuals with COPD and primary care HCPs willing to participate in in-depth interviews | 14 participants -10 patients (range: 50–80 years); 3 (30%) males; 4 HCPs (details not provided) | IDIs were conducted separately for patients and HCPs; field notes were used to record conversations and observations. Data were analysed thematically. |

(*Continued*)

**Table 1.** (Continued)

| Author, year | Study design | Study objectives | Study settings and location(s) | Study population | Participant characteristics* | Data collection and analysis methods |
|---|---|---|---|---|---|---|
| Zeb, 2021 [70] | Qualitative, phenomenological | To understand the lived self-care experiences of patients with COPD and explore the role of the family in self-care | Multi-center, tertiary care settings in Swat, Pakistan (Rural/ semi-urban) | Individuals (>30 years) with COPD patients undergoing treatment and engaged in self-care at home or community | 13 participants; 30–72 years (range); 9 (69.2%) males | IDIs collected patients views of living with COPD, symptoms and their effect on patient's lives, self-care practices and challenges in self-care. Data were analysed using Interpretive analysis based on Ricoeur's interpretation theory. |
| Bhandari P, 2016 [71] | Mixed methods, sequential explanatory | To describe self-care, perceived social support, diabetes management self-efficacy among individuals with T2DM | Multi-center, primary and tertiary care settings in Nepal (Urban) | Individuals (>40years) with T2DM patients attending outpatient clinics | 230 participants (quant) 56.9 ±10.8 years; 91 (39.6%) males 13 participants (qual); age and gender distribution not provided | Self-care behaviours assessed using SDSCA scores. Path analysis identified direct and indirect effects of predictors of self-care behaviour. Qualitative data on barriers and facilitators to self-care were analysed thematically. |
| Jennings H, 2021 [72] | Mixed methods, concurrent | To examine care-seeking practices, management of diabetes, patient experiences of access and quality of care | Multi-center, community-based settings in Faridpur, Bangladesh (Rural) | Individuals (>30years) self-reporting T2DM patients and taking part in the D-MAGIC trial. | 292 participants (quant); 39 (13.2%) 30–39 years, 69 (23.1%) 40–49 years; 81 (30.9%) 50–59 years, 72 (29.4%) 60–69 years, 31 (25.0%) ≥70 years; 146 (50%) males 36 participants (qual) 6 patient in-depth interviews, 50% male, 5 FGDs (5 participants each), 5 in-depth interviews with local health workers | Quantitative surveys, qualitative IDIs and FGDs collected information on diabetes complications, care-seeking, medication, and testing. Logistic regression identified associations between diabetic practices, gender, and socioeconomic wealth quintiles. Qualitative data on care-seeking were analysed using the framework approach. |

*Age and gender details are presented where available. Age is presented in mean years unless specified otherwise.

**Acronyms used:** CVD: Cardiovascular disease; COPD: Chronic Obstructive Pulmonary Disease; T2DM: Type 2 Diabetes Mellitus; EHFSCB: European Heart Failure Self-Care Behaviours; FGD: Focus group discussion; HCPs: Health Care Providers; HF: Heart Failure; IDI: In-Depth Interview; LVEF: Left Ventricular Ejection Fraction; MAQ: Medication Adherence Questionnaire; MMAS: Morisky Medication Adherence Scale; NAFF: Nottingham assessment of Functional Foot Care; NYHA: New York Heart Association; SCHFI: Self-care in Heart Failure Index; SDSCA: Summary of Diabetes Self-Care Activities; SMPQ: Self-Management Practice Questionnaire.

## Indicators of socioeconomic position

Higher age was associated with better self-management in nine studies—the consistency of findings, despite most studies being moderate quality suggested reasonable confidence in the evidence. There was limited evidence of an inverse association from three studies for exercise, self-monitoring and foot care in diabetes [36,41], and for medication adherence in patients undergoing stroke recovery [49]. Two of these studies used higher age cutoffs (≥60, ≥70 years) suggesting that the relationship between age and self-management could be non-linear [53]. Being educated, or having a higher level of education was associated with better self-management in 12 studies. The findings were largely consistent regardless of study quality and the

**Table 2. Observed structural and intermediary-level determinants identified from quantitative and mixed-methods studies, and their reported associations with overall self-management/individual self-management behaviours for selected high burden NCDs in South Asia.**

* Acronyms: COPD: Chronic Obstructive Pulmonary Disease, FC: Foot Care; HF: Heart Failure, HTN: Hypertension, SM: Self-Management (composite), MA: Medication Adherence, T2DM: Type 2 Diabetes Mellitus.

** indicates either singular or multiple self-management behaviours (MA, FC, diet), or self-management (SM) observed using composite scales.

*** Chandrika, 2020 and Gopichandran 2012 report associations with socio-economic status, which includes level of education.

**** Includes family history of disease (Bhandari B, 2015) [34] and lower BMI (Mahmood, 2020 & Gupta, 2022).

Observed association: ◄ = better SM, ▼ = poorer SM, ◄▼ = inconclusive/mixed findings. For studies that observed multiple self-management behaviours, subscripts placed alongside arrows indicate the number of self-management behaviours for which associations were reported.

Sample size: Large arrow ◄ >300 participants; small arrow ◄ <300 participants.

Study quality: Red: Low quality (0–1 point); Amber: Medium quality (2–3 points); green: High quality (>4 points).

wide variation in how the variable was defined—the only study to report an inverse association (i.e. being less educated was associated with higher self-management) was a low-quality study based on a small study sample (n = 150) [74]. Studies that observed multiple self-management behaviours reported mixed findings, with education inversely associated with exercise, diet and foot care [36,48].

Female gender was inversely associated with overall self-management as well as individually observed self-management behaviours in females (n = 8), however higher adherence to medication was reported in four out of six studies, and better foot care in one study. Most studies that assessed gender were moderate to low quality, which limits confidence in the observed associations.

Higher individual or household income was associated with better self-management in six studies, of which four were moderate-quality studies. A limited set of studies, including two that observed wider socio-economic status (incorporating occupation and education in addition to household income) and those observing multiple self-management behaviours reported variations by the type of observed behaviour, with poorer dietary and exercise behaviours reported in higher SES categories [36,39,72].

Occupation was assessed either as employment status (n = 3) or type of employment (n = 2) across five studies. The limited evidence suggests that being employed may be inversely associated with medication adherence as well as overall self-management, while a positive association was observed among those in a higher/professional occupation status.

## Behavioural and biological factors

The presence of comorbidity was associated with better self-management in six studies, of which three studies were of low quality. The observed associations were in studies that either only recorded the presence of comorbidity, or assessed physical comorbidity only. On the other hand, both studies that recorded depression comorbidity reported inverse associations with overall self-management, as well as individually observed self-management behaviours [39,50].

Illness-related factors such as longer illness duration and better control over illness were generally associated with better self-management. While cutoffs for illness duration in most studies ranged from 1–5 years, one study used a 10-year cutoff, reporting better foot care but poorer diet and physical activity among those with a longer illness duration [48]. Another study conducted in rural settings reported poorer adherence to asthma inhalers in those with prolonged illness [52]. Both studies were of low quality. The association between self-management and treatment complexity, the latter being defined as the use of multiple medications or combination treatment, was inconclusive—better self-management was observed in individuals with hypertension while on the other hand, inverse associations were observed in individuals with diabetes.

Smoking was associated with poorer self-management in three out of four studies, the only exception being a low-quality study conducted on asthmatic individuals [52]. Other less commonly observed associations were a lower BMI [41,45] and a positive family history of illness [34] which were associated with better self-management.

## Psychosocial factors

Health-related knowledge was associated with better self-management in all five studies. Despite the limited evidence, which is based on moderate to low-quality evidence, there was consistency in the direction of observed associations, regardless of the type of conditions or the self-management behaviours observed.

Social support was observed as either self-reported presence of support, quality of support, assistance in self-management tasks or living circumstances (i.e., marital status and the type of family system). The presence of social support was consistently associated with better self-management, except for two studies that observed poorer self-management behaviours in joint family settings [38,41]. Despite mostly moderate to low-quality evidence, observed associations were similar across the types of conditions and observed self-management behaviours. In one study, social support was also reported to mediate the association between education status and self-management [71].

Attitudes and beliefs were observed in four moderate-quality studies of individuals with T2DM, hypertension and heart failure. Two of these studies analysed psychosocial constructs derived from the health belief model and theory of planned behaviour. The evidence, although limited, suggests higher levels of self-efficacy and social acceptability to be associated with better self-management. On the other hand, those who required reminders (cues to action) were associated with poor self-management [37,38]. There was either limited or conflicting evidence on other constructs such as for action efficacy (perceived benefits of undertaking self-management) and perceived threat from illness (risk and severity).

## Health-system related factors

Studies exploring health-system related factors were mainly conducted in hypertensive individuals and except for one study, reported associations in relation to medication adherence. Availability of free or low-cost medicine was associated with better self-management, as indicated by four moderate-quality studies—an inverse relationship was reported in one study, but a reasonable explanation was not discernible [53]. Among other determinants, provision of care through qualified providers, treatment satisfaction and regularity of follow-up was generally associated with better self-management. For the latter, one low-quality study reported better adherence to diet and footcare, but poorer physical activity among those with frequent follow-ups.

## Qualitative synthesis

A total of 47 codes were derived from the primary studies, which were organised into eight descriptive themes (S3 Table). The descriptive themes were then interpreted under three cross-cutting analytical themes, supported by illustrative quotes (Table 3).

**Analytic theme 1: The ability to engage in self-management activities is shaped by the degree of financial adversity, available family support, and gender.** This analytical theme (based on descriptive themes 1 and 2) describes a restricted, selective and gendered form of day-to-day self-management, which is underpinned largely by the ripple effects of poverty, reliance on monetary and social support as well as patriarchal socio-cultural norms.

Several studies illustrate how a number of South Asians face crippling adversity in pursuing self-management due to limited household budgets, competing domestic expenses and/or a lack of financial independence [33,55,57,61,63,70–72,75]. As a consequence of this adversity, individuals with NCDs face challenges in the timely and regular purchase of essentials such as medicine and self-monitoring equipment [55,59,60,63–65,70,72] as well as in adhering to recommended dietary regimen [55,56,60,61,63,67] **(quotes 1.1, 1.2)**. Financial hardship also acts as a deterrent to timely health-seeking, especially for rural dwellers and those living on daily wages **(quote 1.3)**. Such individuals routinely avoid the health system to avoid a loss of income, as well as having to pay doctors' consultation fees and associated travel costs [55,58,63,67,70,72,75]. Visits to health facilities are often made only to procure free or

**Table 3. List of illustrative codes from primary qualitative studies supporting the findings of thematic synthesis.**

| Quote no. | Description |
|---|---|
| 1.1 | "The biggest problem is money. If I don't have money, how can I buy medicine? How can I arrange for healthy foods? And how can I go to a health facility, and do blood glucose monitoring?" [55] |
| 1.2 | "You know one thing my doctor told me–just like that on one day–to buy the glucometer to check my sugars at home. He did not ask me whether I can afford it or not. He simply wrote and gave... Already I am spending almost INR 6000–7000 ($84–100) for medication. How can I spend more on my treatment? That machine costs around INR 2000 ($28) and each strip costs around INR 20 ($0.28). Are they really joking?! I won't get these and not ready to get it. Already I am spending almost 1/3rd of my pension on my treatment every month." [59] |
| 1.3 | "It costs 50 taka [approx. 1USD] to go to Faridpur [one way]. It is difficult for a poor person to spend 100 taka [2USD] on a check-up. If a farmer or a day labourer goes to the hospital s/he does not earn their livelihood that day. They also pay for the doctor's bills, tests and medicine" [72] |
| 1.4 | "Without any medical issue, I never visit my doctor. Today I am visiting my doctor after about 6 months. As I believe if I am taking my medicines regularly and without any emergency condition, there is no need to visit my doctor." [61] |
| 1.5 | "I get a lot of support from my family (..) apart from emotional support, my son always gives me money to buy medicines and sometimes he bought medicine from the pharmacy. The support I get from my family motivates me to manage diabetes. (. . .) In our family, we eat the same food, so I don't feel I am eating diabetic diet. I feel lucky to have a caring family." [55] |
| 1.6 | "She (daughter-in-law) often tells me that I am doing drama (pretending to have disease symptoms). She (daughter-in-law) say—If I get support from you (. . ..) you will get food else I don't have time to cook food for you". [54] |
| 1.7 | "There is always tension at home because of expenses incurred due to my medicines. . .24 hours I have to listen to scolding (from husband). . .so, I am fed up. . .I want to discontinue taking medicines so no one will be "allergic" about it." [71] |
| 1.8 | "I was given a course of treatment [at the diabetic hospital] and felt a little better and came back home. I did not go for further check-ups after that as my husband was abroad." Women's FGD [72] |
| 1.9 | "Women in this rural area of Pakistan had a difficult time in managing their diabetes as compared to men. In this society, women cook the food according to the choices of the family–women don't have much to say on the choice of the food, so they have no idea how to manage their diabetes in the environment they live and in relation to the healthy food choices." [57] |
| 2.1 | "The doctor whom I visited was excellent, and he tried his best to make me understand the disease and linkage of tobacco to my condition. Despite this, I was unable to understand either Nepali or Hindi, but, he found some way for me. (Patient laugh) He explained to me by demonstrating by puffing cigarette (moving fingers at his mouth like the smokers) and indicated it causes difficulty in breathing (by touching chest). He was an excellent doctor that I ever visited" [54] |
| 2.2 | "My doctor always encourages me to manage my diabetes at home in each visit. I feel motivated do my best to control diabetes after a visit to a doctor." [55] |
| 2.3 | "I have three diseases (COPD, Hypertension, and Diabetes) but, I do not know either I should take medicine for all disease on a daily basis. To have all the medicines at the same time is really worrisome for me" [54] |
| 2.4 | The doctor has asked me to drink 4 glasses of water daily. I have to take 7 medications twice a day, for my blood pressure, diabetes, cholesterol and abnormal heart rhythm. I need at least 2 glasses of water with them. How can I survive on only 2 glasses of water for the rest of the day?" [75] |
| 2.5 | "My doctor only told me how many times to take the medicines and the pharmacist gave me all the medicines after I paid for them, but nobody ever informed me about the side effects of the medicines and what I should do if I have any problem." [64] |
| 2.6 | "First of all I do not know the steps of exercise, and the next thing is that how exercise will benefit the COPD patients like me. If I were explained both I would have practiced, but I was just told to do exercise, so I ignored it". [54] |
| 2.7 | "I spent nearly 900 rupees on that day, and it is a waste. How much I would have struggled to get (earn) that 900 rupees. Either the breathlessness should go down (sic), or the stomach pain should go down (sic). Both were not happening. Still, it is the same. So that doctor is not good." [67] |
| 2.8 | "The most corruption right now is in the health department. Yes. . .among the doctors. . ..they are taking money that you do not even understand." [72] |

(*Continued*)

**Table 3.** (Continued)

| Quote no. | Description |
|---|---|
| 2.9 | "One person near to my village said to me that he has the medicine for a complete cure for this disease. He provided me with a paste, smelling like ginger and garlic just for Nrs. 100 (0.95 USD). He said, by eating this (paste) thrice a day for a month (three spoon/day) you will get rid of this disease completely. Unfortunately, after eating that (paste), I experienced many side effects involving severe heat production in the body, migraine, and weakness and, eventually, I stopped taking it". (IDI: F70- 75Y) |
| 3.1 | "When patients walk for physical exercise, neighbours ask them- where are you going? Why are you going? Patients had to answer them each time and they feel demotivated to stay active." [55] |
| 3.2 | "At home no issues. If we go out for any family function, then I don't allow him to take insulin. Most of our relatives make that a big issue if they know he takes insulin. Many [relatives] don't go to doctor because doctors have asked them to take [insulin] because of their high diabetes. Moreover, they don't take any treatment at all." (Vinitha, PFM, wife to the patient)" [59] |
| 3.3 | "In Pakistani culture, if unhealthy food is served in parties on a special occasion, it is considered rude not to eat that and bringing diabetes–appropriate food to such events would not be accepted." [57] |
| 3.4 | "Well, I would like to walk every day for more than 30 min, but the roads in my area are not suitable for walking, there is no walkway or park nearby and I am ashamed of doing any exercise in my home" [64] |

subsidised medicine or in unavoidable circumstances such as the deterioration of symptoms **(quote 1.4)**.

The impact of financial adversity can be offset to some extent for those who have available social or monetary support, either from family members or relatives. Social support in particular was seen to confer benefits through knowledge sharing and motivation **(quote 1.5)** [54,55,58,61,65,67,70]—However, many individuals, particularly those who are older or less literate rely on more direct forms of support such as identification and/or administration of medicine or carrying out routine self-monitoring [54,59–61,63,65,67,75]. The availability of such support, however, is not consistent and tends to be lower for women and those living in less educated, less financially stable households **(quote 1.6)**. Women are particularly disadvantaged in this regard—most women in South Asia rely financially on male members of the family to arrange medicine and other self-monitoring equipment, as well as to chaperone them outside the house for exercise or to attend physician appointments [59,61,64,72,75]. The inability of male counterparts to fulfil these roles, either due to poverty or unavailability due to work commitments means that women's self-management needs often remain unmet. Several South Asian women also diminish their own needs, partly due to feelings of guilt (of being a burden) **(quotes 1.7 and 1.8)** and partly to cater to the wishes of family members, such as when it comes to adopting or maintaining a specific diet **(quote 1.9)**.

**Analytic theme 2: Variations in the level and source of health-related information explain differences in the approach to self-management.** This theme (based on descriptive themes 1, 4–5 and 7–8) highlights the importance of health-related knowledge in the South Asian context and discusses how variations in the reported sources of information translate into differences in observed self-management practice.

The role of healthcare providers in supporting self-management is pivotal and complements the individual's sense of responsibility towards oneself and others. In South Asian settings, healthcare providers, particularly doctors are authoritative figures on health-related matters—the advice and positive encouragement received from them can be linked to a good understanding of self-management and its importance, as well as to motivated attempts made by individuals to self-manage illness in day-to-day life **(quotes 2.1, 2.2)** [54,55,61,65,67,72]. Despite its relevance, however, the provision of such support is lacking in many South Asian settings. In general, individuals with NCDs demonstrate insufficient knowledge and confusion

regarding their illness and its appropriate management strategies, which is even more evident in those living with multiple NCDs **(quote 2.3, 2.4)** [54–56,58,61,68,75]. The lack of adequate health-related information is reflected in the fatalistic views held by individuals, [55,58,67] as well as a general lack of interest [54,55,58,59] and can be explained by the nature of their engagement with healthcare providers. In South Asian settings, some individuals may feel reluctant to discuss their needs openly with healthcare providers out of fear of being blamed—communication and knowledge exchange are also hindered by the limited interaction time available in busy outpatient settings, as well as the provision of vague advice such as that around medication-taking and exercise **(quote 2.5, 2.6)** [54–56,59,61,63–65,75].

Individuals who lack knowledge related to self-management are frequently observed to take a liberal, self-directed approach—they prioritise selected self-management tasks (mainly medication use) over others, taking independent decisions around self-management based on perceived symptoms and treatment benefits [54–56,58,61,68,75]. A commonly observed feature is the reliance on non-medical advice offered by well-wishers such as social acquaintances and extended family members [54,55,59,61,67,72]. The experimentation with traditional and alternative methods (herbal remedies, spiritual healing methods etc.), often based on such advice is commonplace, and is used to both complement and replace ongoing medical treatment. The reasons for such choices are often complex, but they are often in part a consequence of negative experiences with past treatment and/or healthcare providers, scepticism of the overall healthcare system, and the appeal of potentially lower-cost remedies **(quotes 2.7–2.9)**.

**Analytical theme 3: "Effective and routine self-management can be precluded by social and physical environments.".**   The last analytic theme (based on descriptive themes 3 and 6) highlights the stigma and lack of social acceptability associated with chronic illness and its management, and how these issues further compound the challenges already present in the harsh physical environments in which individuals live.

Many South Asians feel a general reluctance to engage in self-management activities publicly, whether it is physical exercise, dietary restriction or taking medication [56,59,64,67]. Exercising, for instance, is associated with a sense of shame, particularly when done in outdoor settings—the stigma, in addition to the lack of safe, suitable and gender-appropriate exercise facilities characterise the social and physical environment in which many South Asians, particularly those in urban settings live (**quote 3.1**). In rural settings, it is long travelling distances that act as physical barriers to timely contact with health providers, as well as the routine purchase of medication. These challenges add another layer of complexity for already vulnerable individuals who live with multiple health problems and their manifestations such as fatigue and reduced mobility.

Activities such as taking medication are avoided as they are inconvenient while at work, and also because they can be viewed as a sign of weakness by others. In many instances, individuals purposely disrupt their dietary or medication routine, feeling that their actions will attract unwanted public attention, cause embarrassment and lead to unwarranted questions/judgements (**quote 3.2**). They feel forced to engage in unhealthy activities either due to the unavailability of healthy choices, particularly in relation to diet, simply in an effort to appease their hosts/visitors (**quote 3.3**) [58,61,63,71,75].

## Discussion

Our review consolidates evidence from a large number of recently published studies conducted in South Asian settings. In our quantitative synthesis, we mapped determinants across the Social Determinants of Health framework, reporting their associations with self-management of high-burden NCDs in South Asia. We found that higher age, education and income

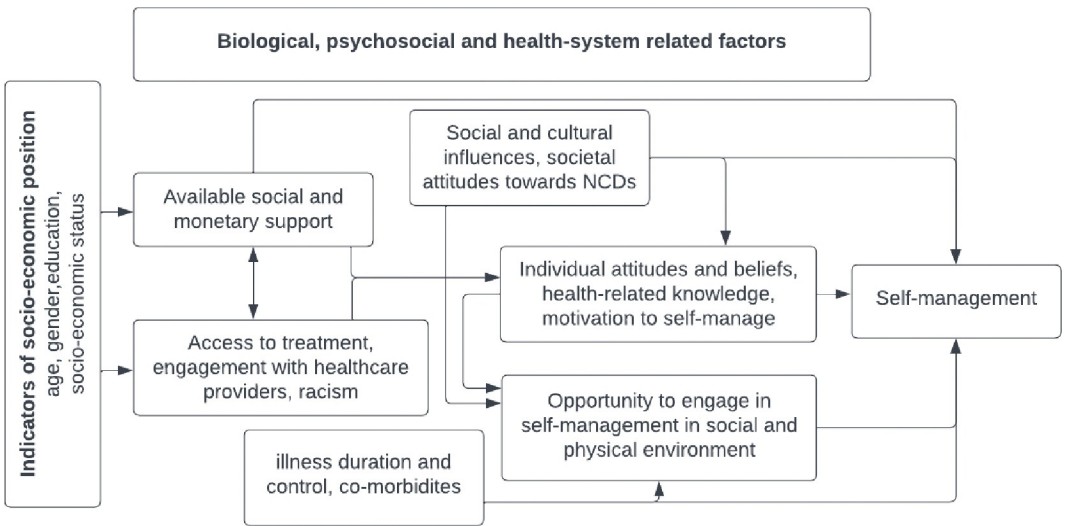

**Fig 2. A conceptual map of self-management in the South Asian context.**

(indicators or socioeconomic position), physical comorbidity, longer duration and better control over illness (biological determinants) better health-related knowledge, self-efficacy and social support (psychosocial determinants), and availability of low-cost, affordable medicine (health-system related determinants) determine better self-management. Our qualitative exploration, conducted in parallel, highlighted how several of these key ingredients were missing in the South Asian context and explained how this affected several self-management activities. The major impediments to self-management among South Asians are the lack of financial resources, poor engagement with healthcare providers and limited health-related knowledge. These challenges in combination with poor social acceptability, reliance on non-medical advice offered by family members/acquaintances and the variation in available social support compounded the challenges in pursuing self-management. Gender-based differences are clearly visible, with women being reliant on male counterparts, burdened with household responsibilities while at the same time less likely to receive family support. In many South Asians, these factors are manifested in the passive and/or self-directed mode of self-management, which is characterised by fatalistic attitudes, poor engagement in self-management activities and/or experimentation with traditional and alternative treatments. Findings from our syntheses clearly highlight the complex interplay of structural and intermediary social determinants in shaping observed self-management behaviours. A conceptual map based on our findings (Fig 2) provides an effective visual representation of these factors and their interrelationships in influencing the self-management of NCDs in South Asian settings.

The findings highlighted above broadly align with and provide an update to the existing evidence on NCD self-management. A qualitative meta-synthesis of studies predominantly carried out in high-and middle-income countries observed that self-management of NCDs is influenced by variations in personal and lifestyle characteristics, health status, available financial and social resources, characteristics of the external physical environment as well as the health system [76]. Our study contextualises several of these features in South Asian settings, using the CSDH framework as a guide. We observe widespread poverty and inequality in South Asia, a feature that is also persistent in wider LMIC settings—our description of the ways it impacts individuals' ability to access timely and necessary care provides a deeper understanding of the differences observed in disease outcomes along the socio-economic

gradient [77]. Our findings also draw parallels with a previously published review focusing on diabetes self-management in wider LMIC populations. This review, which consolidates evidence from studies carried out in Africa, highlights how close-knit family systems can play a vital part in supporting self-management, helping individuals to overcome a range of financial and physical challenges. At the same time, however, it demonstrates how communal practices such as meal preparation and consumption can hinder effective self-management, particularly in relation to dietary regulation [17]. Our review findings complement this evidence, and further add that while family support is an essential ingredient, it is not a guaranteed feature in South Asian settings and can put certain groups such as women and those living in adverse socioeconomic conditions at a disadvantage. We also observed poor social acceptability and perceived shame associated with having a chronic illness, which demotivated individuals and resulted in them forgoing activities such as taking medication and dietary compliance in social gatherings and doing outdoor exercise. Similar social pressures and negative perceptions around NCDs, as well as the lack of health-related knowledge observed in our review have been documented in the South Asian diaspora living in developed, western countries, and explain challenges in adhering to activities such as medication, diet, and physical exercise [14,78,79]. We also found preliminary evidence to suggest that depression co-morbidity results in poorer self-management. Previous research suggests that the type and severity of co-morbidity may be associated with the prioritisation of self-care behaviours for a particular condition [80]. In the case of depression, it could additionally be due to a reduced ability to acquire, retain, recall, and implement knowledge [81].

## Strengths and limitations

The number of South Asian studies on self-management has grown considerably in the past decade, reflecting the interest in understanding and supporting self-management in underlying populations. We used quantitative and qualitative methods to consolidate evidence and provide an overarching synthesis for selected high-burden NCDs. The findings of our review provide new insights in the understanding of health behaviours and challenges related to self-management in South Asian settings; a particular strength is its focus on high-priority marginalised groups such as women and those from a low socio-economic background [82]. Despite these strengths, the review is not without limitations. Firstly, we aimed to identify studies across all South-Asian countries, but despite a comprehensive search we were limited to studies conducted in four South Asian countries (Bangladesh, India, Nepal and Pakistan) only. Our findings may thus have limited generalisability to other South Asian countries which were not represented. We found that overall, studies predominantly focused on self-management of singular conditions, even though in many studies individuals had other comorbidities. There was also a predominance of studies conducted on type 2 diabetes and on the assessment of medication adherence. Quantitative studies were predominantly small-scale and moderate to poor in quality, with limited internal and external validity. Several of these studies relied on non-validated and subjective patient-reported measures to observe self-management behaviours. While the inclusion of these studies enabled us to map a range of determinants, interpretation of some of these may warrant caution, as has been clarified in our narrative synthesis. Qualitative studies rated better overall in quality; despite this, several studies offered only contextually thin data which may have limited the depth, interpretability, and generalisability of our findings. We found that the quantitative evidence focuses largely on indicators of socio-economic position, and less on determinants acting at the intermediary levels, particularly psychosocial determinants and those at the health-system level. There is also a paucity of data on determinants that constitute the wider socioeconomic and political context (policy,

governance, cultural and societal values etc) which are likely to be similar across South Asian countries. We used visual methods (effect direction plots) to supplement narrative synthesis. This approach, while broadly useful, has certain limitations: the plot could only be used to present information on the number (and not the type) of self-management behaviours observed in studies of multiple self-management behaviours. We were also unable to provide information on the direction of association for the individual self-management behaviours in such studies, where the observed associations were discordant. Lastly, we used the MMAT for quality assessment, which assumes equal weights for each quality assessment item. We found that this limits the comparability of quality scores across study designs, particularly when mixed-methods studies are involved. In our review, mixed-methods studies rated higher than quantitative and qualitative studies, despite scoring low on the last item (i.e. adherence to individual quantitative and qualitative quality criteria). Consideration needs to be given to address this limitation in future versions of the MMAT.

### Implications for policy, practice and research

Within the wider socio-political context of South Asia, the self-management of high-burden NCDs continues to be shaped by a complex network of factors at the individual, household/ societal and health system levels, as well as the physical environment. A multi-component, whole-systems approach is therefore needed to address differences in disease risk factors and to support secondary prevention for improving disease outcomes across the social gradient [83]. Empowerment of patients, caregivers and the wider community through knowledge-sharing and education can improve individual and collective attitudes towards NCDs, given the positive attitudes towards health education among South Asian communities [84]. Low-cost, community-based programs in particular have been shown to improve awareness of, and engagement in self-management behaviours, both in South Asian settings and beyond [85,86]. On the other hand, the design and delivery of bespoke behaviour change interventions can target high-risk groups such as those with multimorbidities, who are at a disproportionate risk of adverse health outcomes [87].

Health service delivery in South Asian settings can benefit from communication skills training for healthcare providers, enhanced care coordination, as well as task-shifting initiatives such as the involvement of community health workers [88,89]. Efforts in this direction can contribute towards the delivery of standardised care, improved patient counselling and a stronger emphasis on self-management, which are core components of the World Health Organisation's Package of Essential Non-communicable disease interventions (PEN), [90]. Rural and far-flung areas can particularly benefit from digital and telehealth initiatives, with several small studies demonstrating its potential in South Asia [91]. However, such approaches need to consider issues regarding accessibility, which is a recognised problem particularly among women in rural settings [92]. The involvement of, and engagement with stakeholders in the local context (political actors, health and technology services, end-users) in these initiatives is critical to the sustainability and long-term success of these efforts.

We propose that future research should emphasise the development, implementation and evaluation of the initiatives identified above, while at the same time expanding the knowledge base around self-management, particularly among individuals living with multiple NCDs using objective, validated measures.

### Conclusion

Self-management in South Asia is shaped by financial adversity, lack of health systems support and poor health-related knowledge. A strong social component influences self-management in

these settings- families and close relatives play a pivotal role in the provision of monetary and social support but at the same time societal stigma, patriarchal norms and poor health-related information offsets these benefits. A whole systems approach, using multi-disciplinary, multi-pronged interventions is needed to address gradients in social determinants and improve self-management for high-burden NCDs. Research efforts should focus on the evaluation of such interventions, in order to inform evidence-based practice and policy and extend findings to wider LMIC regions.

## Supporting information

**S1 Table. PRISMA checklist.**
(DOCX)

**S2 Table. Item-specific and overall quality scoring of primary quantitative, qualitative and mixed-methods studies using the MMAT checklist.**
(DOCX)

**S3 Table. List of descriptive themes, sub-themes and individual codes generated in the qualitative thematic synthesis.**
(DOCX)

**S1 Text. Final search strategy.**
(DOCX)

## Acknowledgments

The authors would like to acknowledge the contributions of Professor Catriona McDaid, Professor Richard Holt and Dr Claire Carswell, for supporting the development and conduct of this review in their role as Faraz Siddiqui's PhD advisory panel. Thanks also to the University of York librarian, Mr David Brown, for supporting the development of search strategies.

## Author Contributions

**Conceptualization:** Faraz Siddiqui, Catherine Hewitt, Hannah Jennings, Najma Siddiqi.

**Formal analysis:** Faraz Siddiqui.

**Investigation:** Karen Coales, Laraib Mazhar, Melanie Boeckmann.

**Methodology:** Faraz Siddiqui, Catherine Hewitt, Hannah Jennings, Najma Siddiqi.

**Project administration:** Faraz Siddiqui, Karen Coales, Laraib Mazhar.

**Resources:** Catherine Hewitt, Hannah Jennings, Najma Siddiqi.

**Software:** Faraz Siddiqui, Melanie Boeckmann.

**Supervision:** Catherine Hewitt, Hannah Jennings, Najma Siddiqi.

**Validation:** Karen Coales, Laraib Mazhar, Melanie Boeckmann.

**Writing – original draft:** Faraz Siddiqui.

**Writing – review & editing:** Faraz Siddiqui, Catherine Hewitt, Hannah Jennings, Karen Coales, Laraib Mazhar, Melanie Boeckmann, Najma Siddiqi.

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
