## [Decision Letter · Decision Letter 0]

26 Apr 2023

PGPH-D-23-00207

Self-management of chronic, non-communicable diseases in South Asian settings: A systematic mixed-studies review

Dear Dr. Siddiqui,

Thank you for submitting your manuscript to PLOS Global Public Health. After careful consideration, we feel that it has merit but does not fully meet PLOS Global Public Health’s publication criteria as it currently stands. Therefore, we invite you to submit a revised version of the manuscript that addresses the points raised during the review process.

We look forward to receiving your revised manuscript.

Kind regards,

Nicola L. Hawley

Academic Editor

Journal Requirements:

Additional Editor Comments (if provided):

Reviewers' comments:

Reviewer's Responses to Questions

**Comments to the Author**

1. Does this manuscript meet PLOS Global Public Health’s publication criteria? Is the manuscript technically sound, and do the data support the conclusions? The manuscript must describe methodologically and ethically rigorous research with conclusions that are appropriately drawn based on the data presented.

Reviewer #1: Yes

Reviewer #2: Yes

Reviewer #3: Yes

2. Has the statistical analysis been performed appropriately and rigorously?

Reviewer #1: N/A

Reviewer #2: Yes

Reviewer #3: Yes

3. Have the authors made all data underlying the findings in their manuscript fully available (please refer to the Data Availability Statement at the start of the manuscript PDF file)?

Reviewer #1: Yes

Reviewer #2: Yes

Reviewer #3: No

4. Is the manuscript presented in an intelligible fashion and written in standard English?

Reviewer #1: Yes

Reviewer #2: Yes

Reviewer #3: Yes

5. Review Comments to the Author

Reviewer #1: Thank you for inviting me to peer review the manuscript ‘Self-management of chronic, non-communicable diseases in South Asian settings: A systematic mixed-studies review’. Please find below my comments which I believe if addressed will strengthen this review:

a. Lines 31-33 – Please add key databases searched and also the publication dates considered.

b. Line 77 -need to have a full stop before ‘however’.

c. Please be more specific about the context upfront.

d. Lines 93-94 – What is your definition of self-management and what are the specific behaviours (and measures for quantitative/mixed-methods studies) you were interested in?

e. I suggest the authors update their searches given the final search was conducted over a year ago.

f. Lines 134-135 - Please explain why social-political context (eg. conflict) and physical environment (eg, green spaces) were not considered here.

g. Line 137 – suggest replacing ‘they’ with ‘these’.

h. Line 155- suggest replacing 42 ‘titles’ with ‘studies’

i. Figure 2 – Please explain how from 14,302 titles and abstracts (deduplicated 14,279+23) you have reached 1,631 for 1st stage screening, what happened to 12,671 titles and abstracts?

j. Did you exclude any of the studies based on quality? If not, why? Did you consider how the inclusion of poor quality studies (eg, 8 poor quality quantitative studies) have impacted the results?

k. Table 2 - This is a nice table, but suggest reformatting it slightly as not clear for the reader the direction of association between a given determinant and multiple outcomes (2 and more outcomes, what specific outcomes do you refer to here?)

l. Lines 418 – Consider the effects of physical environment and wider political context here.

m. Lines 420-422 – Please provide specific strategies on how to do these effectively in South Asian context; these are not homogeneous contexts.

n. Lines 424-429 – These are very generic statements; I suggest the authors provide further elaborations on (a) future research directions, and (b) how some of the proposed strategies, eg., implementation of package of essential NCD interventions, and others can be implemented considering global actors, context specific local stakeholders and the political environment.

Reviewer #2: This manuscript provides insight into an important area of mitigating the effects of non-communicable diseases (NCDs) with its focus on self-management strategies among individuals living with chronic NCDs. It fills a gap in NCD research associated with self-management strategies—which undeniably are important when one is working to improve the health of individuals with NCDs. The focus examining the social determinants of health including biological, psychosocial and health system factors and what drives self-care is a strength of this manuscript.

This research will serve as the basis for additional qualitative and quantitative research documenting what affects self-care among individuals with NCDs. Further research, as the authors point out should serve as the basis for building programs to mitigate these individuals’ challenges. One of the strengths of this comprehensive review is the inclusion of both qualitative, quantitative, and mixed methods studies. One limitation that ought to be pointed out is the use of self-reported adherence outcomes in most research reviewed—a limitation not of the manuscript—but of the current status of research on self-care—including adherence in the field. This well-written manuscript which needs minor editing for a few typos (e.g. in Figure 2)."

Reviewer #3: I appreciate the opportunity to provide comments on this systematic review of mixed methods by Siddiqui et al. The review addresses an important gap in the literature to better understand the determinants of NCD self-management in the South Asia region. While I am very enthusiastic about the focus of the review, there are a few questions and concerns that I have about the methodology that make it difficult to understand the presented results.

Major

Data availability: There are inconsistencies between the provided statement “All the data underlying our findings are included within the manuscript and supplementary information provided.” and the data in the supplementary materials. For example, the ‘final search strategies’ include ‘Date of search: 1st March 2021’. This information does not align with the presented information in the methodology of the main text. Please check that all data are presented accurately.

In lines 85 – 86 the registered PROSPERO (CRD42021240899) does not appear to match the title and information presented in this manuscript. It appears that the search strategy casted a very wide net to look at many different countries that were not just ‘South Asian’ and it would be helpful to know why the search was conducted in this way.

In lines 105-107 Did you identify any additional studies from the grey literature search and citation chaining? This is not clear in the text, nor in the PRISMA flow chart.

Please be consistent with your language used throughout the manuscript. For example in line 188, ‘indicators of socioeconomic position’ is mentioned first and then in line 203 ‘SES indices’. These are two distinct concepts and should be clarified.

While I appreciate the conceptual map of self-management in Figure 2, it is difficult for me to see how this is designed specifically for the South Asian context – what about racism?

In lines 396 – 397 : considering that “the number of South Asian studies on self-management has grown considerably in the past decade”, it would be important to acknowledge when the literature search was conducted and using the World Bank ‘South Asia’ region definition for screening as limitation of this work. Whether the search was conducted in March 2022 (as stated in the abstract) or March 2021 (in the supplementary materials), this context is necessary to understand to improve the interpretation of the findings.

Minor

Every table, even in the supplementary materials, should be able to stand-alone. Please check and make sure that all necessary information to understand the data in tables are shown as footnotes. For example, “SMSR ED plots” and Table 2 do not define what the acronyms SM and MA mean. It would also be helpful to remind readers what the points mean and how they were calculated to interpret the table data.

6. PLOS authors have the option to publish the peer review history of their article (what does this mean?). If published, this will include your full peer review and any attached files.

**Do you want your identity to be public for this peer review?** For information about this choice, including consent withdrawal, please see our Privacy Policy.

Reviewer #1: No

Reviewer #2: **Yes: **Danuta Kasprzyk

Reviewer #3: No

---

## [Decision Letter · Decision Letter 1]

1 Aug 2023

PGPH-D-23-00207R1

Self-management of chronic, non-communicable diseases in South Asian settings: A systematic mixed-studies review

Dear Dr. Siddiqui,

Thank you for submitting your manuscript to PLOS Global Public Health. After careful consideration, we feel that it has merit but does not fully meet PLOS Global Public Health’s publication criteria as it currently stands. Therefore, we invite you to submit a revised version of the manuscript that addresses the points raised during the review process.

We look forward to receiving your revised manuscript.

Kind regards,

Nicola L. Hawley

Academic Editor

Journal Requirements:

2. Please ensure that Funding Information and Financial Disclosure Statement are matched.

3. In the Funding Information you indicated that no funding was received. Please revise the Funding Information field to reflect funding received.

4. We do not publish any copyright or trademark symbols that usually accompany proprietary names, eg  ©, ®, ™  (e.g. next to drug or reagent names). Please remove all instances of trademark/copyright symbols throughout the text, including ® on page 30.

Additional Editor Comments (if provided):

Reviewers' comments:

Reviewer's Responses to Questions

**Comments to the Author**

1. If the authors have adequately addressed your comments raised in a previous round of review and you feel that this manuscript is now acceptable for publication, you may indicate that here to bypass the “Comments to the Author” section, enter your conflict of interest statement in the “Confidential to Editor” section, and submit your "Accept" recommendation.

Reviewer #1: All comments have been addressed

Reviewer #3: (No Response)

2. Does this manuscript meet PLOS Global Public Health’s publication criteria? Is the manuscript technically sound, and do the data support the conclusions? The manuscript must describe methodologically and ethically rigorous research with conclusions that are appropriately drawn based on the data presented.

Reviewer #1: Yes

Reviewer #3: Yes

3. Has the statistical analysis been performed appropriately and rigorously?

Reviewer #1: N/A

Reviewer #3: Yes

4. Have the authors made all data underlying the findings in their manuscript fully available (please refer to the Data Availability Statement at the start of the manuscript PDF file)?

Reviewer #1: Yes

Reviewer #3: Yes

5. Is the manuscript presented in an intelligible fashion and written in standard English?

Reviewer #1: Yes

Reviewer #3: Yes

6. Review Comments to the Author

Reviewer #1: None

Reviewer #3: Thank you for the opportunity to re-review the manuscript. The manuscript describes factors associated with the self-management of individuals living with one or more chronic, non-communicable diseases (NCDs) in South Asian settings. While I appreciate the authors’ responsiveness to our feedback to re-run the literature search and to update the language in the manuscript, I have a few questions and hope this can be clarified:

MAJOR

Table 2: what does it mean if there are no subscripts next to the arrow? Are there 0 self-management behaviors observed for a particular structural or intermediary-level determinant? It is difficult to understand why this information is highlighted, especially with the very brief mention of Table 2 in line 183. I’d appreciate additional details either in the footnote and/or in the quantitative synthesis of the results section.

While I appreciate the effort to better define the concepts of socioeconomic position and SES in your revision, it is still not clear to me. Especially in Table 2, it’s a bit confusing to see education as a separate category when it can be encompassed within socioeconomic status.

It is interesting that some countries have 0 studies published and included in this review, with the updated literature search—why is this not acknowledged in the manuscript as part of the discussion? What’s the generalizability of the findings and are there limitations to providing a ‘South Asian perspective’ (as mentioned in lines 82 and 406)?

There are an admixture of race and ethnicities living in each South Asian country setting (as defined by the World Bank definition). Considering that racism exists in healthcare and could influence self-management behaviors—could this be included in your Figure 2 conceptual model?

MINOR

Line 51: Is it appropriate to only mention WHO protocols if the World Bank definition for South Asia countries was used to define the countries included in this review? There is no context for this information, and it is not clear to me why this remains a main discussion point.

7. PLOS authors have the option to publish the peer review history of their article (what does this mean?). If published, this will include your full peer review and any attached files.

**Do you want your identity to be public for this peer review?** For information about this choice, including consent withdrawal, please see our Privacy Policy.

Reviewer #1: No

Reviewer #3: No

---

## [Decision Letter · Decision Letter 2]

27 Nov 2023

Self-management of chronic, non-communicable diseases in South Asian settings: A systematic mixed-studies review

PGPH-D-23-00207R2

Dear Faraz Siddiqui,

We are pleased to inform you that your manuscript 'Self-management of chronic, non-communicable diseases in South Asian settings: A systematic mixed-studies review' has been provisionally accepted for publication in PLOS Global Public Health. The acceptance is contingent upon addressing some minor comments outlined below.

Best regards,

Tara Ballav Adhikari, Ph.D.

Guest Editor

Editor's comments:

It is essential to ensure consistency in the use of acronyms and terminology throughout the document. Specifically, please pay attention to maintaining uniformity with terms such as NCDs or NCD, as well as LMICs or LMIC. Additionally, I recommend a thorough review of the manuscript for grammatical accuracy and coherence before it proceeds to the publication stage.

Reviewer Comments (if any, and for reference):

Reviewer's Responses to Questions

**Comments to the Author**

1. If the authors have adequately addressed your comments raised in a previous round of review and you feel that this manuscript is now acceptable for publication, you may indicate that here to bypass the “Comments to the Author” section, enter your conflict of interest statement in the “Confidential to Editor” section, and submit your "Accept" recommendation.

Reviewer #1: All comments have been addressed

2. Does this manuscript meet PLOS Global Public Health’s publication criteria? Is the manuscript technically sound, and do the data support the conclusions? The manuscript must describe methodologically and ethically rigorous research with conclusions that are appropriately drawn based on the data presented.

Reviewer #1: Yes

3. Has the statistical analysis been performed appropriately and rigorously?

Reviewer #1: N/A

4. Have the authors made all data underlying the findings in their manuscript fully available (please refer to the Data Availability Statement at the start of the manuscript PDF file)?

Reviewer #1: Yes

5. Is the manuscript presented in an intelligible fashion and written in standard English?

Reviewer #1: Yes

6. Review Comments to the Author

Reviewer #1: Thank you for the opportunity to review the revised manuscript. I still have a couple of queries. (1) Could the authors explain why "age" is a structural determinant and not biologic? (2) Please re-read your paper and correct grammar, typos and punctuation, including grammar in newly added statements, e.g., (i) Our findings may thus may

have limited generalisability to other South Asian countries which were not represented. (ii) A limited set of studies,

including two that observed wider socio-economic status (incorporating occupation and education in addition to household income) and observing multiple self-management behaviours reported variations by the type of observed behaviour, with poorer dietary and exercise behaviours reported in higher SES categories”.

7. PLOS authors have the option to publish the peer review history of their article (what does this mean?). If published, this will include your full peer review and any attached files.

**Do you want your identity to be public for this peer review?** For information about this choice, including consent withdrawal, please see our Privacy Policy.

Reviewer #1: No
